Discrimination factors of carbon and nitrogen stable isotopes in meerkat feces

Montanari Shaena shae.montanari@gmail.com
School of GeoSciences, University of Edinburgh , Edinburgh , United Kingdom
Grandori Rita
Electronic publication date: 2017 Jun 13
Publication date: 2017
Volume: 5
Electronic Location ID: e3436
Received 2016 Aug 22; Accepted 2017 May 17
Copyright: ©2017 Montanari
Copyright year: 2017
Copyright holder: Montanari
License: This is an open access article distributed under the terms of the Creative Commons Attribution License, which permits unrestricted use, distribution, reproduction and adaptation in any medium and for any purpose provided that it is properly attributed. For attribution, the original author(s), title, publication source (PeerJ) and either DOI or URL of the article must be cited.
License URL: https://creativecommons.org/licenses/by/4.0/

Keywords: Stable isotopes, Ecology, Dietary ecology, Mammalogy, Meerkats, Trophic ecology, Feces

Funding: Royal Society Newton International Fellowship Funding for this work was provided by the Royal Society Newton International Fellowship. The funders had no role in study design, data collection and analysis, decision to publish, or preparation of the manuscript.

==============================
Stable isotope analysis of feces can provide a non-invasive method for tracking the dietary habits of nearly any mammalian species. While fecal samples are often collected for macroscopic and genetic study, stable isotope analysis can also be applied to expand the knowledge of species-specific dietary ecology. It is somewhat unclear how digestion changes the isotope ratios of animals’ diets, so more controlled diet studies are needed. To date, most diet-to-feces controlled stable isotope experiments have been performed on herbivores, so in this study I analyzed the carbon and nitrogen stable isotope ratios in the diet and feces of the meerkat (Suricata suricatta), a small omnivorous mammal. The carbon trophic discrimination factor between diet and feces (Δ13Cfeces) is calculated to be 0.1 ± 1.5‰, which is not significantly different from zero, and in turn, not different than the dietary input. On the other hand, the nitrogen trophic discrimination factor (Δ15Nfeces) is 1.5 ± 1.1‰, which is significantly different from zero, meaning it is different than the average dietary input. Based on data generated in this experiment and a review of the published literature, carbon isotopes of feces characterize diet, while nitrogen isotope ratios of feces are consistently higher than dietary inputs, meaning a discrimination factor needs to be taken into account. The carbon and nitrogen stable isotope values of feces are an excellent snapshot of diet that can be used in concert with other analytical methods to better understand ecology, diets, and habitat use of mammals.

Introduction

Small mammals are often overlooked in favor of larger more charismatic species, but they fill vital roles as ecosystem engineers, prey base, and seed dispersal agents (Huntly & Inouye, 1988; Brown & Heske, 1990; Davidson, Detling & Brown, 2012). They can live in colonies or in large numbers; therefore their plentiful modern and historical remains can provide records of changing environments and shifting ecological conditions (Terry, 2010). Non-invasive monitoring is an ideal way to track changes in modern mammalian communities, as shed hair and feces can provide a substrate for examining population trends, diets, and health of groups using genetic or chemical methods (Crawford, McDonald & Bearhop, 2008; Pompanon et al., 2012; Rodgers & Janečka, 2013). Specifically examining feces is useful, as it can be collected from rare or cryptic species that are hard to monitor and often avoid humans. Also, feces is plentiful and relatively inexpensive to analyze. Traditionally, vertebrate diets have been assessed macroscopically through physical examination of gut contents and feces (e.g., Hermsen, Kerle & Old, 2016). With the advent of affordable high-throughput sequencing, fecal studies are becoming more common for barcoding dietary DNA, which allows for a more complete dietary picture not as easily biased by differing digestibility of food (e.g., Shehzad et al., 2012; Kartzinel et al., 2015). Stable isotope methods can provide a useful complement to these barcoding studies that are growing in popularity. Fecal stable isotopes have been useful for detecting dietary shifts in mammals such as gorillas (Blumenthal et al., 2012), for understanding weaning time in primates (Reitsema, 2012), and as a climate record from bats (Royer et al., 2015), so understanding isotopic discrimination and variability of this material is vital for future research.

Stable isotope ratios of nitrogen (15N/14N, written δ15N) and carbon (13C/12C, written δ13C) are incorporated in animal tissues and excretions following digestion of food products. Changes in stable isotope ratios can elucidate information about food webs, trophic structure, and habitat use (Ben-David & Flaherty, 2012). It is assumed that carbon isotope ratios do not change drastically as they propagate through the food web (DeNiro & Epstein, 1978), and in terrestrial ecosystems, these ratios are generally used to indicate the primary production at the base of the food web. Carbon isotope ratios of plants that use different metabolic pathways (C3, C4, or CAM) are systematically different, and this difference is incorporated into the tissues or byproducts of consumers. On the other hand, δ15N values are known to become enriched as trophic level increases (DeNiro & Epstein, 1981), and differences in δ15N of tissue over time or between populations can be used to discern changes in food webs, habitat, or prey composition (Post, 2002).

To use stable isotopes in dietary ecology, there has to be a comprehensive understanding of the difference in isotope ratios between diet and tissue (or feces); this difference is called the trophic discrimination factor (also fractionation factor or discrimination factor) and is caused by isotopic fractionation during digestion and metabolism. Trophic discrimination factor is denoted as Δ and defined as Δ = δtissue–δdiet (Martínez del Rio et al., 2009). Trophic discrimination factors (abbreviated TDFs) are widely variable among animals and differ depending on species, tissue examined, and diet type and quality (Caut, Angulo & Courchamp, 2009). General TDFs are used in many mixing model studies for animal diet reconstruction, but it has been shown that small differences in TDFs in these types of studies can lead to vastly different conclusions about dietary makeup (Ben-David & Schell, 2001). Experimental work by Caut, Angulo & Courchamp (2008) shows that mixing models are most accurate when species-specific TDFs are obtained.

In this study, I calculate TDFs for carbon and nitrogen isotopes in meerkat (Suricata suricatta) feces in order to obtain fecal TDFs in a small mammalian omnivore. Meerkats are small, diurnal mammals that are members of Carnivora and are herpestids, closely related to mongoose. They have generalist diets and in the wild consume insects, berries, reptiles, and other small invertebrates (Doolan & Macdonald, 1996). The diets of generalist consumers are difficult to study in the wild due to unknown variability in their diet, so a zoo-based study of meerkats in this instance will provide dietary control, allowing for less variation in the determination of a TDF. Studies like this one are necessary, as TDFs are not often determined for terrestrial mammalian omnivores and carnivores or even feces in general (Tables 1 and 2). Even though feces represent a snapshot of diet, it is often collected in sizeable quantities for non-invasive studies. This can provide important ecological and environmental records and elucidate short-term and seasonal dietary variability. Changes in isotope ratios from diet to feces need to be calculated, as there may be isotopic fractionation from the process of digestion and waste excretion.

Table 1 Fecal trophic discrimination factors of carbon (Δ13C) from this study and from the literature.

Average δ13C of diet used to calculate each TDF is included from the original listed publication. All isotope values are presented ±1 standard deviation if it was available in the original publication. Asterisks (*) indicate when one species from the same experiment was divided into different diet treatments. The column “n” represents the number of fecal samples used to calculate the TDF in each experiment. The Δ13C values in bold are statistically significantly different from zero in their original experiments (in some cases this was not specifically tested).

Species	δ13C diet	Δ13C	n	Diet class	Reference	
Peromyscus maniculatus	−19.4 ± 0.3	−3.2	5	Herbivore	Hwang, Millar & Longstaffe (2007)	
Myodes gapperi	−19.4 ± 0.3	−4.2	5	Herbivore	Hwang, Millar & Longstaffe (2007)	
Microtus longicaudus	−19.4 ± 0.3	−2.7	5	Herbivore	Hwang, Millar & Longstaffe (2007)	
Microtus pennsylvanicus	−19.4 ± 0.3	−3.6	5	Herbivore	Hwang, Millar & Longstaffe (2007)	
Tamias amoenus	−19.4 ± 0.3	−3.0	5	Herbivore	Hwang, Millar & Longstaffe (2007)	
Zapus princeps	−19.4 ± 0.3	−5.9	5	Herbivore	Hwang, Millar & Longstaffe (2007)	
Lama glama*	−13.3 ± 0.3	−1.2 ± 0.4	4	Herbivore	Sponheimer et al. (2003a)	
Lama glama*	−27.0 ± 0.2	−0.4 ± 0.5	4	Herbivore	Sponheimer et al. (2003a)	
Capra hircus*	−27.0 ± 0.2	−0.8 ± 0.1	4	Herbivore	Sponheimer et al. (2003a)	
Capra hircus*	−13.3 ± 0.3	−1.0 ± 0.4	4	Herbivore	Sponheimer et al. (2003a)	
Bos taurus*	−13.3 ± 0.3	−0.9 ± 0.2	4	Herbivore	Sponheimer et al. (2003a)	
Bos taurus*	−27.0 ± 0.2	−1.00 ± 0.2	4	Herbivore	Sponheimer et al. (2003a)	
Oryctolagus cuniculus	−27.0 ± 0.2	−0.3 ± 0.1	4	Herbivore	Sponheimer et al. (2003a)	
Vicugna pacos*	−27.0 ± 0.2	−0.4 ± 0.4	4	Herbivore	Sponheimer et al. (2003a)	
Vicugna pacos*	−13.3 ± 0.3	−1.3 ± 0.2	4	Herbivore	Sponheimer et al. (2003a)	
Equus caballus*	−27.0 ± 0.2	−0.5 ± 0.4	4	Herbivore	Sponheimer et al. (2003a)	
Equus caballus*	−13.3 ± 0.3	−0.7 ± 0.2	4	Herbivore	Sponheimer et al. (2003a)	
Uncia uncia	−23.61 ± 3.15	2.30 ± 1.66	10	Carnivore	Montanari & Amato (2015)	
Panthera tigris	−23.61 ± 3.15	1.25 ± 0.62	7	Carnivore	Montanari & Amato (2015)	
Gorilla gorilla	−28.4	0.3	121	Herbivore	Blumenthal et al. (2012)	
Clethrionomys gapperi*	−29.23	−0.51 ± 1.19	11	Herbivore	Sare, Millar & Longstaffe (2005)	
Clethrionomys gapperi*	−25.49	−1.95 ± 1.04	10	Herbivore	Sare, Millar & Longstaffe (2005)	
Clethrionomys gapperi*	−28.22	0.24 ± 1.20	10	Herbivore	Sare, Millar & Longstaffe (2005)	
Myotis myotis*	−24.54 ± 0.76	−0.17 ± 1.10	15	Insectivore	Salvarina et al. (2013)	
Myotis myotis*	−20.50 ± 0.81	−0.25 ± 0.75	21	Insectivore	Salvarina et al. (2013)	
Rhinolophus ferrumequinum*	−24.54 ± 0.76	0	15	Insectivore	Salvarina et al. (2013)	
Rhinolophus ferrumequinum*	−20.50 ± 0.81	0.09 ± 0.39	21	Insectivore	Salvarina et al. (2013)	
Suricata suricatta	−24.9 ± 3.3	0.1 ± 1.5	24	Omnivore	This study	

Table 2 Fecal trophic discrimination factors of nitrogen (Δ15N) from this study and from the literature.

Average δ15N of diet used to calculate each TDF is included from the original listed publication. All isotope values are presented 1 standard deviation if it was available in the original publication. Asterisks (*) indicate when one species from the same experiment was divided into different diet treatments. The column “n” represents the number of fecal samples used to calculate the TDF in each experiment. The Δ15N values in bold are statistically significantly different from zero in their original experiments (in some cases this was not specifically tested).

Species	δ15N diet	Δ15N	n	Diet class	References	
Peromyscus maniculatus	3.6 ± 0.02	2.1	5	Herbivore	Hwang, Millar & Longstaffe (2007)	
Myodes gapperi	3.6 ± 0.02	2.2	5	Herbivore	Hwang, Millar & Longstaffe (2007)	
Microtus longicaudus	3.6 ± 0.02	2.2	5	Herbivore	Hwang, Millar & Longstaffe (2007)	
Microtus pennsylvanicus	3.6 ± 0.02	2.5	5	Herbivore	Hwang, Millar & Longstaffe (2007)	
Tamias amoenus	3.6 ± 0.02	1.4	5	Herbivore	Hwang, Millar & Longstaffe (2007)	
Zapus princeps	3.6 ± 0.02	2.2	5	Herbivore	Hwang, Millar & Longstaffe (2007)	
Lama glama*	0.4	2.9 ± 0.3	4	Herbivore	Sponheimer et al. (2003b)	
Lama glama*	5.8	3.0 ± 0.4	4	Herbivore	Sponheimer et al. (2003b)	
Bos taurus*	0.7	2.0	4	Herbivore	Steele & Daniel (1978)	
Bos taurus*	0.6	1.7	4	Herbivore	Steele & Daniel (1978)	
Equus caballus*	0.4	2.6	Unknown	Herbivore	Sponheimer et al. (2003b)	
Equus caballus*	5.8	3.3	Unknown	Herbivore	Sponheimer et al. (2003b)	
Ovis aries	0.8	3.0	Unknown	Herbivore	Sutoh, Obara & Yoneyama (1993)	
Capra hircus	1.5	3.6	3	Herbivore	Sutoh, Koyama & Yoneyama (1987)	
Uncia uncia	8.95 ± 0.73	2.49 ± 1.30	10	Carnivore	Montanari & Amato (2015)	
Panthera tigris	8.95 ± 0.73	1.57 ± 2.04	7	Carnivore	Montanari & Amato (2015)	
Sus scrofa	4.6 ± 0.3	1.2	3	Omnivore	Sutoh, Koyama & Yoneyama (1987)	
Gorilla gorilla	3.2	0.6	121	Herbivore	Blumenthal et al. (2012)	
Clethrionomys gapperi*	−0.42	1.76 ± 1.26	11	Herbivore	Sare, Millar & Longstaffe (2005)	
Clethrionomys gapperi*	1.45	1.17 ± 1.68	10	Herbivore	Sare, Millar & Longstaffe (2005)	
Clethrionomys gapperi*	4.00	1.27 ± 2.06	10	Herbivore	Sare, Millar & Longstaffe (2005)	
Myotis myotis*	5.31 ± 0.63	1.81 ± 1.28	15	Insectivore	Salvarina et al. (2013)	
Myotis myotis*	12.88 ± 1.16	2.34 ± 2.17	21	Insectivore	Salvarina et al. (2013)	
Rhinolophus ferrumequinum*	5.31 ± 0.63	0.53 ± 0.54	14	Insectivore	Salvarina et al. (2013)	
Rhinolophus ferrumequinum*	12.88 ± 1.16	0.97 ± 0.45	21	Insectivore	Salvarina et al. (2013)	
Suricata suricatta	4.6 ± 1.8	1.5 ± 1.1	24	Omnivore	This study	

Here I calculate trophic discrimination factors for captive meerkats by measuring the isotopic composition of both the diet and the feces. I also describe how fecal stable isotope values vary over short periods of time so that more information can be gleaned about the measured variability in generalist diets during wild studies. Focusing on small mammals with generalist diets will be key to uncovering ecological factors that cause shifts in mammalian biodiversity, as small mammals are accurate recorders of environmental change over a variety of time scales (Barnosky, Hadly & Bell, 2003; Terry, 2010).

Materials and Methods

Diet and feces samples

The meerkats in this study are maintained at the Edinburgh Zoo (Royal Zoological Society Scotland). Fecal samples were taken randomly from an enclosure containing seven adult female meerkats over the course of April 2016. Subsamples of diet were collected once a week for four weeks to account for variability in diet items. Per the information of Edinburgh Zoo animal care workers, the meerkats are fed different combinations of the food items each day, but over the course of a week the amount of each item they eat is roughly equal. Each animal receives the same amount of each food item by weight. A total of 24 fecal samples were collected, along with 2–4 subsamples of each diet item: carrots and apples, horsemeat, dog biscuits, whole frozen small mice, and whole frozen chicks. The above items were used to calculate the trophic discrimination factor from diet to feces. Homogenized bulk muscle with attached skin was sectioned from the whole chicks and mice for analysis.

Stable isotope analysis

Stable isotope analysis was conducted at the Wolfson Laboratory in the School of Geosciences at the University of Edinburgh. The analysis of carbon and nitrogen isotope ratios was performed on a CE Instruments NA2500 Elemental Analyzer and the effluent gas was analyzed for its carbon and nitrogen isotopic ratios using a Thermo Electron Delta+ Advantage stable isotope ratio mass spectrometer. Sediment standard, PACS-2 (δ15N 5.215‰ (Air) and δ13C- 22.228‰(VPDB)) from the National Research Council Canada was used for isotopic analyses. Acetanilide standard (C 71.09% and N 10.36%) was used for elemental compositions. Isotopic data were determined relative to CO2 and N2 reference gases whose mean values are derived from the average value of PACS-2 samples within each daily run.

The standard deviation for five analyses of the PACS-2 standard run over the same time period as the study samples was ±0.07‰ for δ13C (VPDB), and ±0.14‰ for δ15N (Air). Elemental analysis, measuring the percentage of carbon and nitrogen in the samples, had an error of 1% for carbon and 4% for nitrogen. The stable isotope ratios are reported in standard notation and referenced to air for δ15N values and Vienna Pee Dee Belemnite (VPDB) for δ13C values. Ratios are defined as δ = (Rsample∕Rstandard − 1) where R=13C∕12C or 15N/14N.

Samples were prepared first by lyophilization followed by manual crushing to form a homogenized powder for isotope analysis. Feces and diet samples were subsampled and analyzed both with and without lipid extraction treatment. The samples were lipid extracted by immersion in a 2:1 ratio of chloroform/methanol for 12 h using a Soxhlet apparatus. Following lipid extraction, samples were dried in a 50 °C oven for at least 24 h to evaporate any remaining solvent.

Statistics and data analysis

Statistical tests and analyses were performed in R (R Core Team, 2016), and plots were created in ggplot2 (Wickham, 2009). Trophic discrimination factors were calculated using the averages of all subsampled dietary material. TDFs were calculated for both δ13C and δ15N using the aforementioned equation ΔXfeces = δXfeces–δXdiet where ΔXfeces is calculated in per mil (‰). δXdiet is the value of the average of all non-lipid-extracted dietary samples and δXfeces is each individual fecal stable isotope value.

Parametric statistical tests were performed, as the data are normally distributed as shown through a Shapiro–Wilk test (δ13C: W = 0.946, p = 0.057; δ15N: W = 0.976, p = 0.544). Before hypothesis testing, an F-test was performed to see if a t-test for equal or unequal variance (Welch two sample t-test) should be done based on the results. F-tests show the variance was equal for comparisons between lipid-extracted and non-lipid-extracted feces for both δ13C and δ15N (δ13C: F8,23 = 1.409, p = 0.491; δ15N: F8,23 = 0.408, p = 0.191). Variance for comparisons between diet and feces for both δ13C and δ15N are unequal (δ13C: F15,23 = 5.001, p = 0.0006; δ15N: F15,23 = 2.633, p = 0.036). All means are reported ± standard deviation (SD), and all significance is reported for α = 0.05.

To obtain carbon and nitrogen fecal trophic discrimination factors from other studies for comparison to results from this study, a literature search was conducted using Google Scholar and through mining references in other feces trophic discrimination factor papers. These were all of the papers with diet-feces trophic discrimination factors found as of August 2016 searching keywords such as “feces stable isotopes” and “scat stable isotopes”. These values appear in Tables 1 and 2.

Results

A summary of the stable isotope results is presented in Table 3, and a summary of t-tests is reported in Table 4. All raw isotope data can be found in the supplementary information (Data S1).

Table 3 Stable isotope results from meerkat feces and diet samples (δ13C and δ15N).

Means are shown ±1 standard deviation. Stable isotopes are presented in delta notation (δ) and discrimination factors are noted by Δ. All isotope values are presented in per mil (‰). Trophic discrimination factor in bold is statistically significant from zero.

	n	δ13C (‰)	δ13C range	δ15N (‰)	δ15N range	C/N	C%	N%	Δ13C	Δ15N	
Meerkat scat	24	−24.8 ± 1.5	−28.1,−20.7	6.1 ± 1.1	4.4,8.9	6.9 ± 1.6	20.9 ± 11.1	3.2 ± 1.8	0.1 ± 1.5	1.5 ± 1.1	
Chick	2	−26.1 ± 1.2	−27.0,−25.3	4.5 ± 0.03	4.5	3.7 ± 0.3	47.8 ± 0.3	12.9 ± 1.0			
Mouse	2	−23.4 ± 1.7	−24.6,−22.2	5.2 ± 0.6	4.7,5.6	6.1 ± 1.5	51.4 ± 4.7	8.7 ± 1.3			
Horse meat	4	−27.1 ± 1.3	−28.2,−25.3	7.0 ± 1.3	6.1,9.0	3.5 ± 0.04	46.0 ± 1.9	13.3 ± 0.4			
Fruit mix	4	−27.7 ± 1.5	−29.8,−26.8	3.5 ± 1.0	2.8,4.9	88.6 ± 21.8	38.5 ± 1.0	0.5 ± 0.1			
Dog biscuits	4	−20.2 ± 0.5	−20.7,−19.5	2.9 ± 0.5	2.3,3.4	11.4 ± 1.8	42.9 ± 0.7	3.8 ± 0.6			
Total diet	16	−24.9 ± 3.3	−29.8,−19.5	4.6 ± 1.8	2.3,9.0	27.1 ± 38.1	44.2 ± 4.6	7.1 ± 5.5			

A subset of feces and diet samples were lipid extracted to see if this changed the results of the analysis; however, using a t-test no significant difference was found between the δ13C and δ15N of either material between lipid extracted and non-lipid-extracted samples (Table 4). As in my previous study of fecal TDFs (Montanari & Amato, 2015), I have opted to use only non-lipid-extracted materials for sampling, as a meta-analysis of carbon and nitrogen discrimination factors from Caut, Angulo & Courchamp (2009) has shown that lipid extraction has no significant effect on calculated TDFs.

The mean δ13C value for each food item is as follows: chick (n = 2, −26.1 ± 1.2‰), mouse (n = 2, −23.4 ± 1.7‰), horsemeat (n = 4, −27.1 ± 1.3‰), carrots and apple mix (n = 4, −27.7 ± 1.5‰), and dog biscuits (n = 4, −20.2 ± 0.5‰). The average for all diet items (n = 16) is −24.9 ± 3.3‰. Mean δ15N values are: chick (4.5 ± 0.03‰), mouse (5.2 ± 0.6‰), horsemeat (7.0 ± 1.3‰), carrots and apple mix (3.5 ± 1.0‰), and dog biscuits (2.9 ± 0.5‰). The average δ15N for all diet items is 4.6 ± 1.8‰. The mean diet C/N ratio is 27.1 ± 38.1. The mean δ13C value of the feces is −24.8 ± 1.5‰ and the mean δ15N value is 6.1 ± 1.1‰. The C/N ratio is 6.9 ± 1.6. The ranges and variability of δ13C and δ15N for all materials are presented in Table 3 and the variation of fecal isotope values over the month they were collected is presented in Fig. 1. When the fecal samples were placed into bins by weeks they were collected and subjected to an ANOVA, there was no difference between the average weekly scat values over the course of the month (δ13C: F4,19 = 0.539, p = 0.709; δ15N: F4,19 = 0.886, p = 0.491).

Figure 1 Variability in δ13C (A) and δ15N (B) values of meerkat feces over the month of sampling.

Points on the line represented a measured fecal sample. All isotope values are presented in per mil (‰).

Table 4 Results from t-tests (p-value, t, df) comparing δ13C and δ15N means of stable isotope values from lipid and non-lipid-extracted meerkat feces and diet samples.

The t-test used (Welch or equal) was decided by a preliminary F-test to test for equal variances.

Variable 1	Variable 2	Test	p-value	t	df	
δ13C Scat LE	δ13C Scat	t-test equal	0.12	1.59	31	
δ15N Scat LE	δ15N Scat	t-test equal	0.09	-1.74	31	
δ13C Diet LE	δ13C Diet	t-test equal	0.46	0.75	14	
δ15N Diet LE	δ15N Diet	t-test equal	0.79	0.27	14	
δ13C Diet	δ13C Scat	Welch t-test	0.89	−0.14	19.05	
δ15N Diet	δ15N Scat	Welch t-test	0.01	−2.97	22.61	

The δ13C and δ15N values of feces and diet were compared to establish if there is a significant difference between them. In the case of δ13C there is no significant difference (Welch two-sample t-test, t =  − 0.14, df = 19.05, p = 0.89) while for δ15N the means are significantly different (Welch two sample t-test, t =  − 2.97, df = 22.61, p = 0.01). Discrimination factors were calculated by using the average value of all diet items subtracted from each individual feces sample in order to assess variance. Average Δ13C for feces (Δ13Cfeces) is 0.1 ±1.5‰  and Δ15N for feces (Δ15Nfeces) is 1.5 ± 1.1‰, with only the Δ15Nfeces being significantly different than zero.

Discussion

Trophic discrimination factors of meerkat feces

Feces can be an extremely useful substrate for stable isotope analysis for estimating food input, and these data show in meerkats that in relation to carbon isotopes, feces are a fair representation of diet, but nitrogen isotopes of feces undergo isotopic enrichment. Investigations into stable isotope discrimination factors show that herbivore feces are representative of diet in large bodied animals (Sponheimer et al., 2003a) but that in small-bodied herbivores (Hwang, Millar & Longstaffe, 2007) and non-herbivores (Montanari & Amato, 2015) feces undergo enrichment of nitrogen isotopes within the digestive tract. In this study, the trophic discrimination factor of carbon isotopes from diet-to-feces is small and not statistically significant. Similar results for Δ13Cfeces are seen in studies of insectivores (bats, Salvarina et al., 2013) and carnivores (tigers and snow leopards, Montanari & Amato, 2015). To this point, experimental research on diet-to-feces discrimination factors shows most calculated carbon TDFs are non-significant as referenced by the review of published studies in Table 1. More data are needed on mammalian omnivores and carnivores to further assess this pattern.

The means of δ15Ndiet and δ15Nfeces are different (Table 4); therefore the Δ15Nfeces value (1.5 ± 1.1‰) is significantly different than zero. In this study and the other two non-herbivore fecal TDF studies of Salvarina et al. (2013) and Montanari & Amato (2015), Δ15Nfeces is of similar magnitude and also the only TDF that is different than zero; although Δ15Nfeces is only significant in one out of two species examined in each of these studies. Nevertheless, significance of Δ15Nfeces could indicate 15N enrichment occurs during digestion, likely during transit through the gut as is seen in Hwang, Millar & Longstaffe (2007). This study investigated δ15N values at different parts in the digestive tract of voles and other small rodents and found digested material is enriched in 15N in the stomach, intestine, cecum, and colon relative to the diet. It has also been shown the δ15N of mucosal epithelium was higher than the diet input in some parts of the digestive tract of a sheep, which suggests the enrichment in 15N in feces may be due to the presence of endogenous proteins (Sutoh, Obara & Yoneyama, 1993). In general, significant Δ15N values point to some relationship between digestive processes and Δ15N, but the cause is unknown and more experiments comparing 15N enrichment during digestion in herbivores and non-herbivores are needed.

A number of other factors may be also affecting nitrogen flux during digestion and excretion. Different biochemical pathways related to the changes in proteins occurring during digestion (deamination/amination) could be the cause of 15N enrichment (Hwang, Millar & Longstaffe, 2007). Additionally, the presence of microorganisms in the digestive tract could also enrich fecal matter compared to diet (Hwang, Millar & Longstaffe, 2007; Macko et al., 1987). The 15N enrichment could be due to preferential absorption of 14N by the animal during digestion or removal during urine excretion, but experimental data from Sponheimer et al. (2003b) seem to indicate that at least in herbivores, 14N is not preferentially excreted. In Montanari & Amato (2015), we cautioned that more data should be collected to better understand TDFs of carnivores because the physiological mechanisms are still unknown; this study continues in the same vein and reinforces that there is a preferential loss of 14N and/or enrichment in 15N occurring during movement through the gastrointestinal tract in relation to solid waste. Further mammalian physiological experiments may explain the mechanism.

Variables that affect trophic discrimination factors

It has been shown that TDFs vary due to a number of factors, one of them being the initial isotopic ratio of the diet. Significant relationships between both δ13C and δ15N of diets and TDFs have been shown in bears (Hilderbrand et al., 1996; Felicetti et al., 2003), rats (Caut, Angulo & Courchamp, 2008), and birds (Pearson et al., 2003). A meta-analysis of TDFs and dietary isotope ratios values by Caut, Angulo & Courchamp (2009) finds significant negative relationships between these variables in both carbon and nitrogen. A similar study of this relationship with fecal TDFs cannot be completed due to the fact most of the calculated fecal TDFs are not significantly different from 0 (data from Tables 1 and 2). This could be due to the fact that feces represents immediately ingested diet (within hours or days) as opposed to assimilated diet and is not subjected to as many physiological processes of fractionation. This is a promising result for the use of feces in isotope studies because the variability of TDFs due to dietary input is a major issue for stable isotope food web studies. Variability that needs to be accounted for with other tissues (Caut, Angulo & Courchamp, 2009) appears to be a non-issue in feces.

Mammalian body size might influence calculated TDF (Hwang, Millar & Longstaffe, 2007). A mechanism that could explain this is a general trend of higher mass-specific metabolism in animals with smaller body mass (Kleiber’s law), which could in turn impact the fractionation of isotopes that occurs after food ingestion (Pecquerie et al., 2010). Hwang, Millar & Longstaffe (2007) did a meta-analysis of fecal TDFs in the literature combined with their rodent TDFs and found significant differences in Δ13C between different body sizes, but only in herbivores. A lack of published fecal TDFs for non-herbivorous mammals of different sizes means statistical test cannot be performed for herbivores vs. non-herbivores.

Isotopic variability over time

Feces can be used for point estimates of diet, but long term collection is especially useful for finding seasonal patterns in dietary variability (e.g., Blumenthal et al., 2012). Mammalian carnivores and omnivores tend to change their diets seasonally, so it is important to realize the magnitude of fecal isotope variation day-to-day for wild studies (e.g., Kincaid & Cameron, 1982; Melero et al., 2008). Other than this study, Blumenthal et al. (2012) and Salvarina et al. (2013) track stable isotope ratios of animal feces at monthly or daily intervals respectively.

I have tracked meerkat fecal stable isotopes over the course of one month, and it is clear there is variation on any given day a sample is taken (Fig. 1), as the combination of diet items they eat changes daily. There is a range of 7.3‰  in the δ13C and 4.5‰  δ15Nfeces of meerkat feces over a month. This suggests feces can be directly reflective of a highly variable diet, and also shows δ13Cfeces may not be buffered in small mammals as much against day-to-day variability as in larger mammals (Blumenthal et al., 2012). It is not known exactly how long it takes isotopes of feces to reach dietary equilibrium in meerkats, but in Salvarina et al. (2013) it was shown that bat feces acquired a new dietary signal in 2–3 h so it stands to reason it is also within hours for a small mammal. Examining the daily fluctuations in δ13C and δ15N in feces indicates timing of major dietary changes can be pinpointed quite precisely using this method, at least within days. Due to the fact the TDFs in this study were calculated using an average diet value, the TDF is also averaged and is meant to act as a general guide for a TDF in the wild. This variability is important to realize for wild studies, as it emphasizes the need for larger sample sizes of feces, such as multiple samples per day or week, to lessen the impact of day-to-day variability if researchers are seeking a long-term ecological or environmental trend (e.g., Blumenthal et al., 2012).

Conclusions

I found the δ13Cfeces of captive meerkats was not changed compared to the dietary input, while δ15Nfeces is higher than diet. Compared with other published fecal TDFs, the meerkat data fit with observed trophic discrimination factors, and also show that an enrichment of 15N and/or a depletion of 14N is occurring during digestion or in the gut during gastrointestinal transit. When these TDFs are compared to other fecal TDFs in published literature, they are similar in that they are mostly non-significant, which removes one layer of uncertainty when using feces in wild animal studies. Looking at the stable isotope ratios for both carbon and nitrogen over the course of the month, it is clear short-term, near-daily variability in diet can be captured using stable isotope analysis of meerkat feces. Captive studies like this one with more controlled feces and diet collection parameters will hopefully lead to better understanding in other understudied groups, like terrestrial small and medium sized mammals with omnivorous diets, so that stable isotope analysis of feces can become a more common tool in mammalian stable isotope ecology.

Supplemental Information

Data S1 Raw stable isotope data

All isotope data for meerkat scat and diet samples.

Click here for additional data file.

The author thanks the animal care staff at Edinburgh Zoo for collecting the diet and feces samples and facilitating this research project. Thanks are extended to Colin Chilcott and Steve Mowbray at the University of Edinburgh for help preparing and analyzing all samples stable isotope analysis. The author gives many thanks for reviews and comments from Roberto Ambrosini and two other anonymous reviewers that greatly improved this manuscript.

Additional Information and Declarations

Competing Interests

Author Contributions

Data Availability

The author declares that she has no competing interests.

Shaena Montanari conceived and designed the experiments, performed the experiments, analyzed the data, contributed reagents/materials/analysis tools, wrote the paper, prepared figures and/or tables, reviewed drafts of the paper.

The following information was supplied regarding data availability:

The raw data has been supplied as Data S1.

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
