# Peer review of "Discrimination factors of carbon and nitrogen stable isotopes in meerkat feces"

_PeerJ, doi:10.7717/peerj.3436_

## Round 0.1 · original submission · Major Revisions

Dear Dr. Shaena Montanari, please address all the concerns of the three reviewers. I agree with the reviewers that such changes would improve the clarity of your manuscript and robustness of your statistical analysis.

·

Basic reporting

No Comments

Experimental design

Beside results of the experimental work conducted by the author, the paper presents also results of meta-analyses conducted to support some of the conclusions. I think these meta-analyses are important, and add value to the work. However there is almost no methodological detail on them in the current version of the paper. I think these analyses should be explained in full details. I list here some of the points that I think should be reported.
When the literature search was conducted? On which databases? Which keywords were used? Was the literature search exhaustive? How many papers were included in each analysis? How many measures? Were data reported in different units standardized? How? Did you account for non-independence of data reported in the same paper or by the same research group in the analysis?

Validity of the findings

No Comments

Comments for the author

I list here my minor remarks:
Abstract L32: I think you should add the scientific name of the meerkat.
Methods L118: Please state for how many weeks you collected samples? Please also state precisely how you calculated the mean isotope content of diet items.
L151 Here you state that the food items you collected make up equal parts of the meerkat diet. This information is important because it justifies your approach to the analyses. I think you should highlight this point when you describe diet samples collection. In addition, you should report whether meerkats are fed by a mix of food items in each day (as it appears from the data reported in the supplementary materials) or by different food items in different days.
L159 (and also in the legend to table) I think this should be “equal or unequal variance”.
Results L183 and elsewhere: please report degrees of freedom for all tests.
L192 and L 283 I am unclear with this analysis. Since you used ANOVA, I think you classified species in different groups according to body size, but you did not provide any information on how body size groups were defined. Please explain.
The Raw stable isotope data file in the supplementary materials is unformatted and it took me some time to understand its structure. Please provide a formatted datafile.

Reviewer 2 ·

Basic reporting

The structure of the manuscript has to be improved. Some parts are placed in the wrong place. For example, after having read the methods you think you know what the author did, but then you have some surprises in the results with some tests not mentioned so far. And later in the discussion another surprise: variability over time which is not presented at all in the results. Also in the ‘statistics’ part actually there are no statistics mentioned.

Experimental design

The experimental design could have been better. One main withdraw of the study is that the isotope values of the feces varied a lot within the one month of the experiment (lines 247-248: There is a range of 7.33‰ 248 in the δ13Cfeces and 4.5‰ δ15Nfeces of meerkat feces over a month) but the same diet isotopic values were used. Although the species is omnivorous it might not feed equally on all available food items provided. Also there might be individual differences that are totally ignored. These can lead to miscalculation of the discrimination factor. If the feces analysed are some from only meat diet and some or another day from mixed diet or only fruits, the diet isotopic values you use are always the same, which might lead to bias in the calculation of the trophic discrimination. I wouldn’t raise this issue if instead of feces there was a tissue with long turn over rate. But feces reflect the diet of the last day (or even hours in bats) or last few days.
The sample size is rather small, 7 samples per week for 4 weeks and those are randomly collected from 7 individuals. It’s possible that you took feces from the same individual the same day. Captive animals in zoo are excellent opportunities for well-designed experiments but I don’t see this chance was used. Also the number of diet samples is not very high, it’s one every week (if not every two weeks).
I don’t know how precise these discrimination factor values are. Of course these values are better than nothing (as there are no other information for meerkats) or better than using values calculated for other animals and since the results agree with those of other studies on insectivorous small mammals it is highly possible that the results are indeed trusted and useful. The points raised above, can be improved either by collecting additional samples (with known diet the day the feces are collected) or with further discussing some methodological issues raised below in the specific comments (e.g. whether the animals were eating the same proportions of all food items each day etc).

Validity of the findings

Some points related to this are mentioned in the 'experimental design' session.

Comments for the author

I think some more care should be paid on reporting the results from other papers (see in the specific comments more).

Nice introduction.

It’s nice you summarize in Tables 1 and 2 your results together with results from literature. These tables could contain more information, such as the isotope value of the feces and information about the specimens if available, e.g. sex. Also the sample size should be mentioned.

The structure of the manuscript has to be improved. Some parts are placed in the wrong place. For example, after having read the methods you think you know what the author did, but then you have some surprises in the results with some tests not mentioned so far. And later in the discussion another surprise: variability over time. Also in the ‘statistics’ part actually there are no statistics mentioned.

Specific comments

No need to repeat the word ‘significantly’ all the time. Since you talk with statistical results if there is difference it is implied it is significant.

Abstract, Line 39: what exactly do you mean ‘there is no effect of isotopic ratio of diet on discrimination factor in feces’? so no matter of what is the isotopic ratio of the diet there is no effect on the discrimination factor? Apparently this is not a conclusion from your experiment because you didn’t test what effect have different isotopic ratios of food on the discrimination factor. Is this a conclusion from the literature study? Or the whole sentence is just not well written? (“Unlike discrimination…………… in feces,” the rest is fine).

Line 53: by using your previous reference Montanari & Amato (2015) here it seems like you took samples from animal in the wild and that you showed in this study that these rare species are hard to monitor and avoid humans, while in your study you took samples from animals in a zoo!

Line 73: different isotope values (δ13C and δ15N) can also relate to different habitats (marine or aquatic in general vs terrestrial)

Line 81: You could give in brackets the other terms that are used instead (e.g. fractionation factor).

Line 94: …due to unknown…

Lines 98-99: decide if you want to use feces in plural or singular, here in the same sentence you use them in both ways.

Line 96: maybe a comma is needed after the TDFs? Please check for the correct use of commas in the whole text. I had the impression that some commas might be needed in some cases.

Lines 105-106: this is something you actually did not do in the results

Line 122: respectively? If not, you used both types of samples? Why skin? This seems an unusual choice. I don’t know of many studies comparing skin and other tissues isotopes. I hope skin’s isotope is representative of the whole body’s isotope values since the mice and chicks you were using probably come from cultures that have the same diet.

Line 152: equal parts in what? Volume/weight?

Statistics and Data Analysis: I think you have to first compare the fecal samples of each sampling date and then merge them

You give a lot of statistics and methods in the results section while they should be better placed in the ‘Statistics and data analysis’ part. Actually in ‘statistics and data analysis part’ there is no specific statistical tests mentioned at all! Instead you give information (line 151) that I would put in the ‘diet and feces samples’ part

Line 151: were the meerkats consuming all the available food? Is each meerkat eating each day equal proportions of all food available? Do they get each day all food items? I think these are important points that need to be answered in order to know if the discrimination factors you calculated make sense.

Line 189: do you mean you put them all in the same regression (1-2 data points for each species) Or you checked per species using raw data?

Line 201: in which isotopes? Also why herbivore? In this sentence you talk about non-herbivorous as well.

Line 206: why ‘other’ insectivores? I don’t think you mentioned any insectivores so far.

Line 207: in line 203 you talk about enrichment and now about not significant enrichment. What is it true or for which animals what?

Line 208: you repeat in different words what you said before in lines 205-206. Better make this one sentence.

Line 215: in Montanari and Amato you write in the conclusions ‘the nitrogen TDFs
do not seem to be significant and could potentially be ignored (except in the case of snow leopard scat)’ but here you generalize this conclusion. The same with the study Salvarina et al.

Line 216: Where this value comes from? (1.47±1.51‰)

Line 216: ‘the only TDF’ sounds like TDF were measured for 10 isotopes and only for N it was significant. Actually only in Salvarina et al. 3 isotopes were used (from which the TDF for N was significant for the 1 of the 2 species only). In Montanari & Amato there are 2 isotopes.

Line 227: is a word missing here? may are also affecting

Line 228: I think something is missing so that the sentence makes sense, commas and/or the word ‘occuring during’

Line 234: in Montanari and Amato you cautioned that more data should be collected for carnivores (not omnivores).

Line 241: time was ignored so far…

Line 244: add the word respectively and put the words ‘daily and monthly’ in the right order

Line 247: it’s surprising that we read about this only now in the discussion. I think this variation is important and can influence the results.

Line 248: the word feces in subscript is not necessary because you mention the word in the sentence

Line 251: what do you mean here? At least (there must be other studies as well) in the diet shift experiment in Salvarina et al. it was calculated that in 2 insectivorous bat species the feces acquired already the new dietary isotopic diet in 2-3 hours.

Line 265: the variables are just diet isotopes

Line 275: Salvarina et al. didn’t study the effect of body size but they mention another study that did.

Line 287: I don’t think this ‘however’ is needed there.

Line 299: ‘like this one’ I would suggest ‘improved compared to this one’, with known diet at the day of the feces collection, with higher sample size and with feces collected from specific animals.

Table 1: Some species are presented twice with different values each time. What is the difference (e.g. males vs. females, juveniles vs. adults, different diet, time?) or is it a typo? It would be useful to note the differing factor e.g. with a * and a note under the table (or in the legend).

Table 1: check the values please.
For example, in Salvarina et al. the Δ13C are both positive for R. ferrumequinum but you give one negative.


Table 1 & 2: it would be good to write next to δ13C for diet the word ‘diet’ because someone can be confused and think it’s the tissue/feces value.

Reviewer 3 ·

Basic reporting

The article is written in English, and I have no major comments on language. I have some minor comments on the text (detailed in General Comments). Introduction and background is sufficient. Figures are OK. Data is summarized in Tables 1-3, but an additional table is needed with all raw data.

Experimental design

The submission describes original research with a clearly defined and justified question. The methods are clearly described and replicable.

Validity of the findings

Data collection, analysis, and presentation are appropriate. The conclusions are appropriately stated and supported by the data.

Comments for the author

This study presents data on diet-feces isotopic offsets using a captive (zoo) population of meerkats. The results resemble previous studies on mammals. I have no major comments on the study design, analysis, or interpretation, although I do have some comments that I would like to see addressed before publication.


Line 53: “rare or cryptic”

Line 54: “it” is ambiguous. specify what you mean

Lines 54-55: Please provide a reference for example of macroscopic investigations of diet. Additionally, vertebrate diets are also studied by behavioral observation. If the author specifically means indirect methods for assessing diet then yes physical examination of gut contents has been the conventional approach.

Lines 55-58: Might be useful to say that the combination of DNA barcoding AND stable isotope analysis represents a fruitful approach (e.g. Kartzinel et al DNA metabarcoding illuminates dietary niche partitioning by African large herbivores. PNAS 112:8019-8024).

Lines 104-105: “Here I calculate trophic…. both the diet and the feces.” This sentence is repetitive, please delete.

Lines 108-109: “Trophic discrimination factors… ecological and environmental trends.” This sentence is repetitive, please delete.

Lines 109-111: Can you provide a few references to studies showing that small mammal ecology is important for understanding mammal biodiversity etc?

Results: please only report isotope values to one tenth per mill. Reported values to the hundredth implies greater precision than there is.

Line 158: Please specify what criteria were used to determine if the data are normally distributed. Is this based on a statistical test (e.g. graphic or statistical test)?

Lines 199-201: “Feces can be…. must be better understood.” I would delete this sentence, as this was already established in the introduction. Better to start with a statement on what you found in this study, then put it into context.

Lines 201-204: “Investigations into herbivore…. within the digestive tract”. Specify which isotopes, because if you are referring to carbon then that would contradict what you say later in the paragraph. First you say that feces DO undergo enrichment in non-herbivores, and then you say that similar results (lack of enrichment) are found in invectives and carnivores. This is confusing. It might be more effective to use one sentence at the end of the paragraph comparing the results presented here for a small-bodied omnivorous mammal to previous studies of herbivores and carnivores. Do meerkats look more like the herbivores or carnivores, in terms of diet-feces discrimination?

Lines 207-209: Delete the sentence “For meerkats, d13Cdiet and d13Cfeces are not significantly different, indicating d13Cfeces is a fair approximation for diet”, which essentially repeats a previous sentence (Lines 204-206).

Lines 209-211: The use of the word “trend” seems odd. It would be better to simply say that in most taxa diet-feces offsets are small. Also, since the majority of studies focus on herbivores, I’m not sure how useful this kind of statement is anyway. Might be more effective to generalize what each diet class looks like rather than across all studies.

Lines 213-239: Could you provide more details as to the possible aspects of digestive physiology in non-herbivores vs herbivores that may explain the difference.

Lines 217: “likely during transit through the gut” This is more likely than what other possibilities? Why is this possibility more likely? Please explain, and provide references.

Discussion section “Isotopic variability over time” - i would move this section to the end of the discussion. It is strange that this section on temporal variability is sandwiched between sections addressing TDFs. Additionally, is there evidence from behavioral observation studies that small omnivorous mammals have highly variable diets (seasonally, or otherwise)? This would demonstrate why this technique may be useful.

Lines 245-247: I agree the figure and calculated ranges of d13C and d15N values indicate a lot of variation. But what does it mean that there “is significant variation on any given day a sample is taken”. Is there a statistical test you are referring to? What is “significant”? Don’t use that word in this context unless you are referring to a statistical result. It might be useful to calculate the average difference between successive samples, although it won’t be possible to test whether or not each value is statistically different from each other since the groups would have sample sizes of 1.

Lines 248-250: Do you have daily collections? Just because your samples are isotopically variable, that doesn’t mean they are not buffered. Are you capturing the full isotopic range of the constituent foods? Additionally, (putative) differences in buffering between small and large mammals could be due to differences in gut passage time. Has this been studied in meerkats or similar small omnivores?

Lines 250-251: “It is not known exactly how long it takes isotopes of feces to reach dietary equilibrium” No not exactly, but there are studies of mammal gut passage time in the literature, and a few diet-switch experiments where isotopic equilibration time can be estimated. But yes, clearly fecal equilibration should not pose a problem for investigating monthly-scale variability.

Line 254: what do you mean by “large sample sizes” - large sample volumes? repeated collections on daily/weekly etc intervals?

---

## Round 0.2 · Major Revisions

PeerJ has a strict policy on manuscripts containing statistical analyses. The reviewers have provided major methodological input ,which you should satisfy to meat the technical standard required for publication.

·

Basic reporting

No Comments

Experimental design

No Comments

Validity of the findings

I found this new version of the paper improved and much easier to read than the previous one. The author also modified the text according to most of my comments. Unfortunately, there is still one important methodological aspects of meta-analysis that was totally overlooked during the revision. I am referring in particular to my previous comment on how the author accounted for non-independence of data in the meta-analyses (“Did you account for non-independence of data reported in the same paper or by the same research group in the analysis?”). I found no specific answer to this point. The author stated that regression analysis and ANOVA was used, but these analyses did not account for non-independence of data, which is a major issue in meta-analyses, because results from the same study are less independent to one another than those from different studies (indeed, they share the same methods, the same people making the lab and the statistical analyses etc.). The results of meta-analyses can be seriously biased if non-independence of data is not accounted for.
There is also a second important issue with meta-analyses, which I missed in my previous comments. The author apparently did not account for the variance of the results from different study. For instance, in this study, delta-15N is estimated to be 4.6±1.8‰ (by the way, the author should state whether 1.8 is the standard deviation or the standard error), but then only the value 4.6 is used in the meta-analysis. I guess that all the other studies reported standard error/deviation for delta-N and delta-C values, but they were omitted from the meta-analysis. This is a large and unnecessary loss of information because meta-analytic procedures allow accounting for variance in the results.
As far as I can see from the tables in the manuscript, the meta analyses should be conducted by using mixed models, with study as a random grouping factor and diet or body size as a random covariate within study. Each delta value should also be weighed by the reverse of its variance. However, the author should consider these my advices as a very rough guidance, as more problems might be hidden in the data. For example, data from the same species are non-independent. However from the tables it appears that each species was investigated in one study only, so in this case study should account for both variability among studies and among species. Phylogenesis might/should also be considered. I therefore strongly suggest discussing in details the analyses a statistician with an expertise in meta analyses. I can also suggest the excellent paper by Van den Noortgate et al (Three-level meta-analysis of dependent effect sizes. Behav Res 45:576–594, 2013) for methodological details on meta-analyses.

Comments for the author

Minor comments
Line 45. I would add a short statement on why small mammals are often overlooked. I guess because they are often elusive.
Line 49. It is a bit unclear what the author exactly means with "baselines"? General conditions?
Line 81. I would add here, rather than below, that the discrimination factor is also called trophic discrimination factor. I think that adding this further definition later adds confusion.
Line 161. How did you graphically examine data to assess normality of distribution? Did you use qq-plots? Please specify. I would also suggest using Levene’s test instead of F-test to assess equality of variance, because the F-test is known to be extremely sensitive to non-normality.
Line 207 and elsewhere. Please, report degrees of freedom for all F statistics.
Line 211 and elsewhere. Please, report also the residuals degrees of freedom for all F statistics.
Table 4. In the table (not in the legend) please use t-test and Welch t-test instead of t-test equal and t-test unequal.

Reviewer 2 ·

Basic reporting

Figure and table legends should be improved so that they contain all necessary information.

Experimental design

No comments.

Validity of the findings

I’m not very happy about how the main result of the experiment is presented. I think there has been effort to present it as more general as it is. Although some comments have been added concerning daily variability in the feces isotopes the fact that the discrimination value that is calculated is an average value for feces of omnivorous animals using an average of food items (and not necessarily the food items they had consumed the day before) and not necessarily the correct discrimination value for any random feces of that animal is not admitted. This would be important if someone wants to track changes in diet in short scales.

Comments for the author

Title: I would add something that shows that there is some literature review as well.

Line 33: trophic discrimination of what?
Also in the brackets there should be information that can be missed without impact on the content, here you don’t explicitly say it’s carbon and then later there is a Δ15N while you haven’t said that Δ is trophic discrimination factor but it seems as ‘Δ13C’ is trophic discrimination carbon.

Line 37: ‘more directly’ compared to what?

Line 39: ‘A meta-analysis shows that unlike discrimination factors from other tissues and materials’: Rephrase. Your meta-analysis did not deal with other tissues so I wouldn’t start the sentence with ‘a meta-analysis shows…..’ because it could be confusing and someone might think that your metanalysis was bigger that it actually was. Also which other materials you refer to here? Normally the samples can be tissues or feces.

Line 40: in which isotopes?

For example Lines 52-53 and 59-60: citations can be in random order? Not alphabetically or chronologically?

Line 82: why don’t you give here all synonyms of trophic discrimination and the TDF?

Line 85 for example: feces is not a tissue and here it’s talking about tissues (so not for feces). In these cases I would suggest the use of the word ‘sample’

Lines 167 and on: how did you compare between lipid and non-lipid extracted samples? It’s not clearly said.

Line 169: ‘results’: which results exactly?

Line 204: did you compare (with a statistical test) between the 4 weeks?

Line 284: ‘it appears no such relationship with feces exist’ you just said that in before in the previous sentence

Line 301: Did you do this test also between groups or for each group separately? (omnivores, carnivores and insectivores)

Lines 317-318: Does this variability have to do with the size or the diet? Since they are omnivores, a variability should be expected no matter of the size I would say.

Line 330-331: here you always talk about the average of feces and the average of the diet. I find not precise your statement. If you had taken one feces produced from the real food item that was consumed the day before you don’t know what the difference would be.

Line 333-334: ‘or by the addition of microorganisms in the gut during gastrointestinal transit’: how did you results show this? To me it seems unsolicited comment following the ‘also show’

Delta that is δ is not explained anywhere but suddenly ‘delta’ is used

Line 150: all feces and diet samples were subsampled? I find this unnecessary. You could put the effort, time and money instead to analyse more feces.

Line 219: ‘body mass’ redundant, if you say 5-100kg should be enough

Lines 229-239: it is not clear when you talk about nitrogen and when about carbon. There doesn’t seem to be an order. E.g. line 229 which isotope?

Line 237: this is in mammals, not general

Line 241: it looks like you repeat something you said in the previous paragraph

Line 243: it is Salvarina et al. (2013) not Salvarina (2013)

Figure 1: a fullstop is missing after the word ‘sampling’. You could mention that it is feces of meerkat.

The legends in general are not informative enough. They should stand alone and provide all information needed to understand what they show. Please also mention the word ‘meerkat’ and feces whenever needed.

Table 4: if someone reads only this table legend has no idea what it is about.

Why do you use sometimes scat and sometimes feces?

Line 575: Doesn’t the scientific name need to be in italics?

Table 1: your reply to my previous comment was ‘Perhaps I am misunderstanding but Table 1 in Salvarina et al. shows the ‘light labeled’ diet Δ13C for R. ferrumequinum to be negative. That is where I obtained this value.’
Please check again table 1 and figure. δ13C for R. ferrumequinum feces is the same as the diet (light labeled diet).

---

## Round 0.3 · Minor Revisions

I ask you to satisfy the remaining minor concerns and requests of additions/clarifications raised by the reviewer

·

Basic reporting

I found this version much improved. The papers fulfils all the requests of the journal. I have only few minor suggestions listed below.

Experimental design

no comment

Validity of the findings

no comment

Comments for the author

I list here my suggestion for further minor improvements to the text.

Abstract L34: I am not sure your statements that carbon discrimination factor is “…not significantly different from the dietary input” is the most straightforward ways to express this concept. I would say that 0.1±1.5 is not significantly different from zero. You can then explicitly comment this result with “that means that carbon stable isotope ratio in the faeces is not significantly different from the dietary input” or something like that. Please modify also the part on nitrogen (L36) accordingly.

L141 I think your statement that “Elemental analysis error was 1% for carbon and 4% for nitrogen” might be misunderstood. Admittedly, I am not an expert in isotope analyses, however, when I read this sentence I was impressed by the size of the error (1%) compared to that of the isotope ratios analysed, which is in the order of per mill. My you please to clarify this point?

L165 and in all the following: please round numbers to the third decimal at most. I also suggest using the following notation (example from line 165) F8,23 = 1.409, P = 0.491 or alternatively F = 1.409, df = 8,23, P = 0.491. Please note that F8,23 should be written with 8,23 in subscript, but I could not do that in this form. Please also use P (either capital or low case) or p-value consistently throughout the manuscript.

L171-176 I am rather sure that a reader who do not know previous version of this paper will not understand why you did a literature search and report all values in Tables wihout formally analysing them. I think you should briefly explain here why you did this review and why it is important. Similarly, since now you do not report a meta-analysis in the paper, I don’t think you need to justify why you did not include results in a meta-analysis (see e.g. L 278-280). I think a reader unaware of the previous versions of this paper will be surprised by these statements.

L179 repeats L 168. I suggest removing L179.

L201 and 2012 denominator df are missing here.

L206 Please report t, df and P in this order.

Tables 1 and 2: why not reporting also SEs for delta-C and delta-N values?

Table 4: I think the last column is not necessary and should be removed.

---

## Round 0.4 · accepted · Accept

The submitted revision of your manuscript satisfies all the requests of minor changes and is now suitable for publication on PeerJ.